# Gender and socio-economic stratification of ultra-processed and deep-fried food consumption among rural adolescents: A cross-sectional study from Bangladesh

Mohammad Redwanul Islam[1]*, Syed Moshfiqur Rahman[1,2], Md. Monjur Rahman[2], Jesmin Pervin[2], Anisur Rahman[1,2], Eva-Charlotte Ekström[1]

1 Department of Women's and Children's Health, Uppsala University, Uppsala, Sweden, 2 International Center for Diarrheal Disease Research, Bangladesh (icddr,b), Dhaka, Bangladesh

* mohammadredwanul.islam@kbh.uu.se

## Abstract

### Background

Although consumption of ultra-processed and deep-fried foods among adolescents is a global health concern, little is known about its gender and socio-economic stratification in rural settings of low- and middle-income countries. We, thus, aimed to describe ultra-processed and deep-fried food consumption among rural adolescents by gender and socio-economic factors, and to explore their relative importance in shaping consumption.

### Methods

This cross-sectional study drew on data from a household survey in Matlab, a rural sub-district in Bangladesh. The analytic sample comprised 2463 adolescents. We assessed consumption of four ultra-processed food groups: ready-to-eat or "instant" foods; confectionery, sweets and similar packaged products; savory snacks; sugar-sweetened beverage; and of deep-fried foods with a 24-hour, qualitative recall. Asset scores were constructed. Proportion of consumption was calculated and compared by gender and household wealth. Logistic regression models were fitted to isolate socio-demographic variables associated with consumption.

### Results

Approximately 83% (81.5–84.4) adolescents consumed at least one ultra-processed or deep-fried item. Confectioneries were the most consumed (53.5%), whereas sugar-sweetened beverage was the least consumed (12%) group. Boys had greater odds of consumption than girls for all food groups. The association was strongest for sugar-sweetened beverage (adjusted odds ratio = 2.57; 95% CI: 1.97, 3.37), followed by deep-fried foods (adjusted odds ratio = 1.96; 95% CI: 1.66, 2.32) and ready-to-eat foods (adjusted odds ratio = 1.85; 95% CI: 1.45, 2.38). Belonging to the richest households was associated with ready-to-eat food consumption (adjusted odds ratio = 1.55; 95% CI: 1.12, 2.16). Adolescents with

**Data Availability Statement:** The study under consideration availed data from the 15-year follow-up of the MINIMat (Maternal and Infant Nutrition

Intervention in Matlab) trial. The 15-year follow-up was a large, collaborative project involving Uppsala University, Karolinska Institute, Finnish Institute for Health and Welfare and International Centre for Diarrhoeal Disease Research, Bangladesh (icddr,b). icddr,b is the local collaborator and ethics approval was obtained from the Ethical Review Committee at icddr,b in Dhaka, Bangladesh. Because of the statutory requirements, internal data policies and regulations existing in the collaborating bodies along with the over-arching General Data Protection Regulation (GDPR), the data must be stored in institutional repository (storage platforms) and cannot be made directly accessible without a review of the request for access to data. Data availability is further limited because the data contain information on gender and health-related and behavioral attributes, and thus, considered to be "sensitive personal data" as per GDPR. While the data are pseudonymized in accordance with GDPR, supplementary information that can link the data to each study participant exist and are preserved following regulations in place at the collaborating bodies. Therefore, the data can be accessed only upon formal request that details the purpose of such request. The request will then be processed by the Data Repository Committee (DRC) at icddr,b (contact: aahmed@icddrb.org). Any such request should be directed to the principal investigators of the MINIMat15y project: Professor Eva-Charlotte Ekström (email: lotta.ekstrom@kbh.uu.se) and Dr Anisur Rahman (email: arahman@icddrb.org).

**Funding:** This research was funded by Swedish Research Council (VR medicin och hälsa, https://www.vr.se/english.html), grant number #2016-01880; acquired by E-CE and AR. The funder had no role in study design, data collection and analysis, decision to publish, or preparation of the manuscript.

**Competing interests:** The authors have declared that no competing interests exist.

higher educational attainment had lower odds of consuming sugar-sweetened beverage (adjusted odds ratio = 0.73; 95% CI: 0.54, 0.98).

## Conclusion

Consumption of packaged confectioneries, savory snacks, and deep-fried foods appeared common, while SSB consumption was relatively low. Role of gender was pre-eminent as consumption was more likely among boys across the food groups. This may disproportionately expose them to the risk of diet-related non-communicable diseases.

## Introduction

Ultra-processed foods (UPFs) refer to multi-ingredient, industrial formulations composed of such food-derived substances as starch, fat, oil, sugar, casein, etcetera; or synthesized through complex processing of food constituents like whey, gluten, soya protein isolate, maltodextrin, and corn syrup [1, 2]. UPFs contribute to suboptimal diets, a global health concern that accounted for an estimated 11 million deaths and 255 million disability-adjusted life-years (DALYs) among adults in 2017 [3]. Suboptimal diets typically have a higher share of refined sugar, added salt and *trans*-fat with a decreasing proportion of whole grains, fruits and vegetables, nuts and seeds [4]. These attributes are related with a global shift away from diets based on homemade meals to those dominated by energy-dense, highly processed food and drink products [1].

Monteiro and collaborators proposed the NOVA (non-acronym) classification in 2010 and coined the term UPF. The word "ultra-processed" signifies the multitude of industrial processes required to generate these hyper-palatable, highly profitable, packaged products containing little to no whole food [1, 5, 6]. Consumption of UPFs has escalated across the world with their dietary share reaching as high as 57% of the daily energy intake in some settings [7]. After dominating the food systems in high-income countries, availability of mass-produced UPFs is rapidly growing in low- and middle-income countries (LMICs) [8]. Epidemiological studies consistently link UPF consumption with greater energy intake; higher consumption of refined sugar, added salt, saturated and *trans*-fats, and lowering of diet quality [9–14]. Unsurprisingly, observational research has associated UPFs with obesity [15–18], heightened non-communicable disease (NCD) risk [8, 19–21], and all-cause mortality [22].

Deep-fried foods represent another category of unhealthy foods. Oil frying increases energy density and improves flavor profile of foods [23]. Owing to palatability and affordability, deep-fried foods are popular among adolescents and often consumed together with ultra-processed products like sugar-sweetened beverage (SSB). A major increase in vegetable oil production in the developing world [24] underlies the wide availability of deep-fried foods. Energy density of deep-fried foods drives increased energy intake. Accordingly, prospective studies capture a positive association of frequent consumption of deep-fried foods with obesity and type 2 diabetes [25]. Deep frying also leads to formation of oil degradation products that have been linked to NCDs including cancer [26]. Nonetheless, policy responses to minimize consumption of these unhealthy foods remain patchy in LMICs [27].

Consumption of ultra-processed and deep-fried foods among adolescents in LMICs is a multi-dimensional challenge. Changes in food environment accompanying nutrition transition have occurred at an unprecedented pace in LMICs [24]. With growth of UPF sales stalling in high-income countries, Asian LMICs are observing marked increase in UPF availability

[28]. Stunting, thinness, and food insecurity affect a significant proportion of adolescents in these settings [29]. Consequently, access to inexpensive calories through these foods may expose them to a disproportionately high NCD risk in ensuing adulthood [30, 31]. Moreover, adolescents are particularly susceptible to highly charged ideas and images employed in pervasive marketing of UPFs through electronic and social media [32, 33]. Evidence also suggests that energy intake among adolescents in LMICs has spiralled, whereas their diet quality remained poor [34]. While the trend of adolescent overweight and obesity has plateaued in high-income countries, it has accelerated in South Asia [35]. Similar to other parts of South Asia, Bangladesh is also experiencing a growing burden of overweight and obesity among adolescents [36]. Ultra-processed and deep-fried foods are known dietary drivers of increasing overweight and obesity [15, 18]. Nevertheless, socio-demographic correlates of consumption of these unhealthy foods among rural, Bangladeshi adolescents remain unexplored. Hence, we aimed to describe ultra-processed and deep-fried food consumption among rural adolescents by gender and socio-economic factors, and to explore their relative importance in shaping consumption.

## Methods

### Participants, data collection and study site

This cross-sectional study utilized data collected during the 15-year follow-up of MINIMat trial (Maternal and Infant Nutrition Interventions in Matlab, reg#ISRCTN16581394). MINIMat was a factorial, randomized trial that recruited 4436 pregnant women from Matlab, between November 2001 and October 2003 to test the effects of food and micronutrient supplementation for pregnant women. The sampling, interventions and results have been reported elsewhere [37]. The cohort of children born to the participating mothers has been repeatedly followed up [38]. The latest, 15-year follow-up was conducted from September 2017 to June 2019. Eligible adolescents (n = 3267) were tracked using unique identification numbers allotted during the trial. Trained enumerators interviewed the adolescents and their mothers/guardians with a pre-tested, structured questionnaire.

Matlab is a low-lying, rural sub-district that is part of Chandpur district. The International Center for Diarrheal Disease Research, Bangladesh (icddr,b) has been running a Health and Demographic Surveillance System (HDSS) in Matlab since 1966. The HDSS currently covers more than 230,000 people in 142 villages [39]. The community is agrarian and rice farming is the commonest occupation.

### Assessment of ultra-processed and deep-fried food consumption

NOVA classifies foods based on the extent and purpose of industrial processing into four categories: (i) natural or minimally processed, (ii) processed culinary ingredients, (iii) processed, and (iv) ultra-processed. Operationalization of this classification and identifying characteristics of UPFs have been published extensively [1, 2, 6, 17]. Common UPFs include biscuits or cookies, chips (crisps) and various savoury or sweet, packaged snacks; ice cream, chocolate and candies; carbonated and energy drinks; "instant" noodles and soup; packaged breads, buns and muffins; frozen meals among others. UPFs available in Matlab were identified in accordance with NOVA and listed systematically during the formative phase of the 15-year follow-up. As recommended by Monteiro et al. [2], we considered industrial formulation and presence of multiple ingredients (typically >5) to identify the UPFs. The locally available UPFs were grouped as follows: (i) ready-to-eat or "instant" foods, (ii) confectionery, sweets and similar packaged products, (iii) savory snacks, and (iv) SSBs (including energy drinks). Foods that

were submerged in extremely hot oil during the cooking process were considered deep-fried. S1 Table shows the grouping of ultra-processed and deep-fried foods.

Consumption was assessed at individual level through a single, qualitative recall of all ultra-processed and deep-fried foods consumed in the past 24 hours of the survey [40]. Reported consumption of roughly one tablespoonful or more of one or more items qualified the participant as consumer of that particular group of ultra-processed or deep-fried food [41]. The enumerators probed for unreported consumption after the initial recall using list and pictorial aid for minimizing recall bias. The pictorial aid displayed photographs of ultra-processed and deep-fried foods available in Matlab, listed and taken during the extensive formative phase of the 15-year follow-up.

## Socio-demographic variables

Socio-demographic variables included in the analysis were gender, household wealth, maternal and adolescents' education. Gender was a dichotomous variable (girl/boy). Household wealth was ascertained with asset score [42]; computed for each household from principal component analysis (PCA) of ownership of a set of durable assets (e.g., mobile phone, radio, television, refrigerator, bicycle, etcetera), access to electricity and sanitary latrine, and nature of fuel used. PCA yields factor scores for each variable and eigenvectors for the principal components. The asset score variable, with a mean of zero and a standard deviation of one, was created using the factor scores for variables within the first principal component as weights. Asset scores arranged in ascending order were divided into tertiles with the lowest, intermediate and highest representing the poorest, the middle-status and the richest households, respectively. Educational status was categorized according to completed years of formal education: none, primary (1–5 years), and secondary (6–12 years) for adolescents or secondary and above ($\geq 6$ years) for mothers. Because of the low number of adolescents in the category of primary education (n = 62), we combined this category with that of no education during analysis.

## Statistical analysis

All analyses were performed in R, version 3.6.3 (The R Foundation, Vienna, Austria). Sample characteristics are presented as frequency with percentage or mean with standard deviation (SD). We calculated consumption proportion across categories of gender and household wealth, and differences between groups were tested with Pearson´s Chi-squared Test. We fitted binary logistic regression models to explore association of socio-demographic variables with consumption of ultra-processed and deep-fried foods. The multivariable models adjusted for all four socio-demographic variables simultaneously. Adjusted odds ratios (AOR) with 95% confidence intervals (CI) are reported. Visual examination of the plots of quantile residuals and Hosmer-Lemeshow Test were done to assess adequacy of the models. Variance inflation factors (VIF) were retrieved, and none exceeded 2.5. Two-sided P-values $<0\cdot05$ and 95% CIs that exclude 1 were considered statistically significant.

## Ethics statement

This study was conducted in strict accordance with the Declaration of Helsinki. The Ethical Review Committee at icddr,b in Dhaka, Bangladesh, approved the 15-year follow-up of MINI-Mat trial (reference: PR–17029). Written informed consent and assent were obtained from the participating mothers and adolescents, respectively; after full disclosure of the purpose, methods, risks, and benefits of the study.

# Results

Adolescents born to MINIMat mothers as singletons with valid birth anthropometrics were eligible to participate. Out of 3267 eligible adolescents, 2465 (75.45%) completed the household survey of 15-year follow-up. The reasons for loss to follow-up were: outmigration (n = 656), child death (n = 94), and refusal to participate (n = 52). After exclusion of two participants for missing data, the analytic sample comprised 2463 adolescents. Table 1 presents their socio-demographic, household and basic anthropometric attributes. Girls constituted slightly more than half of the sample (51.2%), and participants' mean age was 15.03 years (standard deviation 0.16). The vast majority of adolescents had secondary education (80.1%), whereas 44.6% of the mothers attained similar or higher educational status. Most of the households accessed drinking water from tube-wells (92.9%) and had electricity coverage (87.9%). Approximately 52% of the households owned farming land.

Table 2 shows the consumption proportions for four groups of UPF and for deep-fried food by gender and household wealth. About 83% of the adolescents reportedly consumed at least one ultra-processed or deep-fried food in the past 24 hours. Confectionery, sweets and similar packaged products was the most consumed UPF group (53.5%, 51.5–55.4), while SSB was the least consumed (12%, 10.7–13.2). Deep-fried foods were consumed by 41.3% adolescents. Consumption proportion was higher among boys than girls for all five food groups with the highest difference observed for deep-fried food (49.7 *versus* 33.4%, P < 0.001), followed by SSB (17.1 *versus* 7.0%, P < 0.001). While 10.8% of those from the poorest households consumed ready-to-eat or "instant" foods, this increased to 13.1% and 15.1% among their peers

**Table 1. Socio-demographic, household and basic anthropometric characteristics of the adolescents in analytic sample (N = 2463).**

| Characteristics | n (%) or mean (SD) |
|---|---|
| Age in years | 15.03 (0.16) |
| Height in centimeters | 156.52 (7.61) |
| Weight in kilograms | 45.25 (8.99) |
| Gender: | |
| Girls | 1261 (51.2) |
| Boys | 1202 (48.8) |
| Adolescents' education (years of formal education): | |
| None | 428 (17.4) |
| Primary (≤ 5 y) | 62 (2.5) |
| Secondary (6–12 y) | 1973 (80.1) |
| Maternal education (years of formal education): | |
| None | 491 (19.9) |
| Primary (1–5 y) | 873 (35.4) |
| Secondary and above (≥ 6y) | 1099 (44.6) |
| Household source of drinking water: | |
| Piped water | 147 (5.9) |
| Tube-well water | 2287 (92.9) |
| Others (rain or surface water) | 29 (1.2) |
| Household electricity coverage | 2166 (87.9) |
| Farming land ownership | 1290 (52.3) |

The values represent frequency (n) with percentage for categorical variables and mean with standard deviation (SD) for numerical variables.

**Table 2. Prevalence of consumption of ultra-processed and deep-fried foods among the adolescents by gender and wealth strata.**

| Categories | Ready-to-eat or "instant" foods | Confectionery, sweets, and similar packaged products | Savory snacks | Sugar-sweetened beverages | Deep-fried foods | One item from any of the five groups |
|---|---|---|---|---|---|---|
| | n (% of total in that gender or wealth category, 95% CI) | | | | | |
| **Overall (n = 2463)** | 320 (13.0, 11.7–14.3) | 1317 (53.5, 51.5–55.4) | 872 (35.4, 33.5–37.3) | 295 (12.0, 10.7–13.2) | 1018 (41.3, 39.4–43.3) | 2043 (82.9, 81.5–84.4) |
| **Gender:** | | | | | | |
| Girl (n = 1261) | 118 (9.4, 7.8–10.9) | 644 (51.1, 48.3–53.8) | 421 (33.4, 30.8–35.9) | 89 (7.0, 5.6–8.4) | 421 (33.4, 30.8–36.0) | 1005 (79.7, 77.5–81.9) |
| Boy (n = 1202) | 202 (16.8, 14.7–18.9) | 673 (55.9, 53.2–58.8) | 451 (37.5, 34.8–40.3) | 206 (17.1, 15.0–19.3) | 597 (49.7, 46.8–52.5) | 1038 (86.3, 84.4–88.3) |
| **P-value[a]** | < 0.001* | 0.014* | 0.032* | <0.001* | <0.001* | < 0.001* |
| **Household wealth:** | | | | | | |
| Poorest (n = 822) | 89 (10.8, 8.7–12.9) | 416 (50.6, 47.2–54.0) | 289 (35.2, 31.9–38.4) | 82 (9.9, 7.9–12.0) | 324 (39.4, 36.1–42.7) | 642 (78.1, 75.3–80.9) |
| Middle-status (n = 819) | 107 (13.1, 10.8–15.4) | 438 (53.5, 50.1–56.9) | 270 (32.9, 29.7–36.2) | 100 (12.2, 9.9–14.4) | 340 (41.5, 38.1–44.9) | 689 (84.1, 81.6–86.6) |
| Richest (n = 822) | 124 (15.1, 12.6–17.5) | 463 (56.3, 52.9–59.7) | 313 (38.1, 34.8–41.4) | 113 (13.7, 11.3–16.1) | 354 (43.1, 39.7–46.4) | 712 (86.6, 84.3–88.9) |
| **P-value[a]** | 0.037* | 0.067 | 0.095 | 0.061 | 0.321 | < 0.001* |

Row percentages with 95% confidence intervals (CI) are presented in parentheses.

[a]Based on Pearson's Chi-squared Test.

*Asterisk indicates statistical significance at P < 0.05.

from the middle-status and richest households, respectively (P = 0.037). Proportion of adolescents consuming at least one item from any of the five groups increased along household wealth: from 78.1% among those from the poorest to 86.6% among those from the richest households (Table 2).

Table 3 presents the adjusted odds ratios from multivariable binary logistic regression analyses that accounted for gender, household wealth, maternal and adolescents´ education simultaneously. Gender was associated with ultra-processed and deep-fried food consumption. The odds of reported consumption of all four groups of UPF and of deep-fried food were higher among boys compared to girls. The most pronounced gender difference was observed for SSB (adjusted odds ratio (AOR) for boys = 2.57; 95% CI: 1.97, 3.37). Boys also had 1.85 times higher odds of consuming ready-to-eat or "instant" foods (AOR = 1.85; 95% CI: 1.45, 2.38). The odds of deep-fried food consumption were nearly two times higher among boys (AOR = 1.96; 95% CI: 1.66, 2.32).

Household wealth was associated with the consumption of ready-to-eat or "instant" foods and SSB, but not with consumption of the other three groups. Compared to adolescents from the poorest households, those from the richest households had significantly higher odds of consuming ready-to-eat or "instant" foods (AOR = 1.55; 95% CI: 1.12, 2.16) and of consuming SSB (AOR = 1.44; 95% CI: 1.02, 2.03). Maternal education was not associated with consumption of any of the five food groups (odds ratios not shown). Adolescents' educational status was inversely associated with SSB consumption with the odds being lower among those with secondary education compared to those with lower educational attainment (AOR = 0.73; 95% CI: 0.54, 0.98). All P-values from Hosmer-Lemeshow Test were >0.05, excluding inadequate fit (Table 3). The complete regression analyses with both the crude and adjusted ORs can be found in S2 Table.

## Discussion

The objective of the present study was to describe and analyze the gender and socio-economic stratification of ultra-processed and deep-fried food consumption in a cohort of rural adolescents. The results showed that proportion of adolescents consuming the four groups of UPF and the deep-fried foods ranged from 12% to 53.5%. Packaged confectioneries were the most commonly consumed food group, followed by deep-fried foods and savory snacks. Consumption of SSB and ready-to-eat or "instant" foods was fairly low. While prevalence of consumption varied by gender for all food groups, it varied by wealth status only for ready-to-eat or "instant" foods. Boys were more likely to consume ultra-processed and deep-fried foods than girls. Household wealth was positively associated with consumption of ready-to-eat or "instant" foods and SSB, whereas educational status of adolescents was inversely associated with SSB consumption.

The majority of the adolescents reported consumption of packaged confectioneries and similar products. Matlab is a rural area without super- and hyper-markets and transnational fast food chains. Instead, small-scale, independent retailers are the common distribution channel for UPFs in Matlab. They operate from small stores–rural equivalent of "corner stores" in high-income, urban settings [43]–mostly selling cheap UPFs available in small packets. Price of these match the affordability of rural consumers or scant pocket money that the adolescents receive from their parents [44]. This may explain the highest consumption proportion observed for packaged confectioneries and similar products. Evidence suggests that these alternative retailers play a critical role in expanding sales of UPFs in low-income, Asian settings [28]. These mass-produced, packaged confectioneries are loaded with refined carbohydrate, added sugar, salt and saturated fat, and additives, and serve as important source of inexpensive calories [1]. More than 40% of the participants consumed deep-fried foods. Household wealth status did not predict consumption of deep-fried foods in our study. We could not ascertain whether this reflected uniform affordability of deep-fried foods across wealth categories, or

**Table 3. Logistic regression analyses of association between socio-demographic variables and consumption of ultra-processed and deep-fried foods.**

| Variables | Ready-to-eat or "instant" foods | Confectionery, sweets, and similar packaged products | Savory snacks | Sugar-sweetened beverage | Deep-fried foods |
|---|---|---|---|---|---|
| | Adjusted[a] odds ratios (95% CI) | | | | |
| **Gender:** | | | | | |
| Girl (Ref.) | | | | | |
| Boy | 1.85 (1.45–2.38)* | 1.22 (1.04–1.44)* | 1.19 (1.01–1.41)* | 2.57 (1.97–3.37)* | 1.96 (1.66–2.32)* |
| **Household wealth:** | | | | | |
| Poorest (Ref.) | | | | | |
| Middle-status | 1.25 (0.92–1.72) | 1.08 (0.89–1.33) | 0.91 (0.74–1.13) | 1.21 (0.88–1.68) | 1.08 (0.88–1.33) |
| Richest | 1.55 (1.12–2.16)* | 1.19 (0.96–1.48) | 1.17 (0.93–1.46) | 1.44 (1.02–2.03)* | 1.21 (0.97–1.51) |
| **Adolescents' education:** | | | | | |
| Primary and below (Ref.) | | | | | |
| Secondary | 0.78 (0.58–1.04) | 1.13 (0.92–1.39) | 0.97 (0.79–1.21) | 0.73 (0.54–0.98)* | 0.99 (0.80–1.22) |
| **Hosmer-Lemeshow Test** | P = 0.876 | P = 0.941 | P = 0.349 | P = 0.945 | P = 0.715 |

CI: confidence interval; Ref.: reference category.

[a] Adjusted for gender, household wealth, maternal and adolescents' education simultaneously.

*Asterisk indicates statistical significance as CI does not include 1.

existing perception of "instant" items and SSB being expensive and extravagant compared to deep-fried foods. Moreover, there may have been behavioral factors [45] influencing deep-fried food consumption–meal skipping, snacking, eating away from home, screen time, etcet-era–that we could not capture. Deep frying is considered a health-compromising way of food preparation [46]. Such aspects as type of oil used, type of food being fried, and frying mode (deep *versus* pan frying) are relevant in this regard [25]. Cost largely determines the type of fry-ing oil used in low-income settings and repeated use of frying oil is a common practice [26]. Palm and soybean oils are commonly used for frying in Bangladesh [47]. Palm oil contains 45–50% saturated fatty acids and concentration of trans fat in partially hydrogenated soybean oil can reach up to 20% [25]. Consumption of deep-fried foods, thereby, may predispose the participants to obesity.

Consumption of SSB appeared low among MINIMat adolescents. Studies demonstrate that SSB consumption is more prevalent in urban than rural settings [48, 49]. In an urban sample of 475 Bangladeshi participants (age range: 18–25 years), Muktadir and colleagues found self-reported SSB consumption to be 78.9% [50]. Prevalence was higher in other urban settings as well: 88.7% in the preceding month of survey in Quebec (n = 311) [51], and 71.4–79.8% in the preceding week of survey in Guatemala City (n = 1042) [52]. Similarly, contrasting with ready-to-eat items' marked contribution to overall UPF intake in high-income settings [12, 13], we observed low consumption of ready-to-eat preparations. Low consumption of these two UPF groups in our study could be an indication of limited purchasing ability. The positive associa-tion between wealth status and consumption of ready-to-eat foods and SSB suggests that ado-lescents from relatively wealthier families in Matlab can afford these products. Contrary to this finding, lower socio-economic status aligns with higher UPF consumption in some settings [53–56]. A couple of nation-wide surveys from high-income countries, however, reveals no association of wealth status with UPF consumption [57, 58].

Qualitative research from Matlab suggests that mothers perceive empty-calorie, UPFs like "instant" noodles, mass-produced biscuits and muffins available in sealed packets to be health-ier alternatives to unhygienic street foods [44]. This is accompanied by a lack of understanding of the inherently unbalanced nutrient profile of UPFs and negative impact of industrial pro-cessing on dietary pattern [44]. Additionally, we demonstrated limited dietary diversity with a monotonous diet dominated by rice and lower consumption of fruits, vegetables and animal-source foods in the same cohort [59]. UPF consumption concurrent with such marginal diet quality, labelled "double burden of suboptimal diet", has been documented in a low-income, rural setting elsewhere [60]. Nonetheless, its health implication among adolescents remains to be studied. Consumption of ultra-processed and deep-fried foods has been associated with sev-eral clinical outcomes including metabolic syndrome, hypertension, dyslipidemia and coro-nary artery disease [25, 61]. These associations were observed predominantly among adults, yet the underlying cardio-metabolic alterations may commence during adolescence [62]. Besides, the habit of consuming unhealthy foods established during adolescence can track into adulthood [63]. Accordingly, the gender stratification observed in this study might play a role in exposing the boys to a higher cardio-metabolic risk in upcoming adulthood.

Gender difference in UPF consumption appears variable: some studies reporting higher consumption among men and boys [54, 57], some documenting the opposite [64, 65], and oth-ers finding no association [10–12]. Although prevalence of SSB consumption was low in our analysis, SSB consumption showed a clear gender stratification. The predominance of boys as consumers of SSB agrees with earlier studies [14, 66, 67] and the qualitative study conducted in Matlab [44]. Gender association in the present study may have resulted from taste prefer-ence, gender norms allowing boys greater access than girls to retailers or vendors as boys stay outdoors more [44], and avoidance of UPFs by girls because of perceiving those fattening [68].

Nevertheless, examination of these themes was beyond the scope of the study. We found an inverse association between adolescents' education and consumption of SSB but no other food groups. This contradicts the positive association of UPF consumption and education shown in a Brazilian cohort, albeit from an urban setting with widespread UPF availability [64]. A recent study among adults of NutriNet-Santé cohort also reports those in the highest quartile of UPF consumption having higher educational status [69]. How adolescents with different educational attainment interact with the food environment in Matlab needs to be explored to understand this discrepant association.

Certain limitations of this study need to be considered for proper interpretation of the findings. We did not objectively ascertain the extent of industrial processing and few items might have been misclassified. Despite trained enumerators adopting a two-pronged approach–free recall followed by probing with list and pictorial aid–that has been recommended [70], recall bias is unavoidable to some extent. Self-reported consumption is also prone to social desirability bias. While adolescents' tendency to under-report consumption of sweet and savory snacks and beverages in 24-hour recall has been documented in LMICs, a recent validation study shows the degree of under-estimation to be acceptable [71]. The findings are generalizable to adolescents in Matlab, given the area-wide recruitment protocol adopted in the initial trial [37]. The findings can also be extended to adolescents in other rural, agrarian settings in Bangladesh taking into account the similarity in socio-cultural context.

## Conclusion

We explored ultra-processed and deep-fried food consumption in a rural birth cohort, and observed a strong influence of gender in shaping consumption. Ultra-processed confectioneries and deep-fried foods were commonly consumed. Widespread consumption of these empty-calorie foods is of concern from a public health nutrition perspective. Boys had a greater likelihood of consumption across the food groups. This may disproportionately expose them to the risk of diet-related NCDs. Household wealth was associated with consumption of ready-to-eat or "instant" foods and SSB, but not the other three food groups. This may indicate a role of purchasing ability in the rural setting. Public health initiatives for promoting healthy diet among adolescents should take into account the gendered consumption of unhealthy foods. Strategies to improve adolescents' understanding of the health-compromising features of ultra-processed and deep-fried foods are also needed.

## Supporting information

**S1 Table. Food items belonging to the four groups of ultra-processed foods and one group of deep-fried foods for which consumption was assessed during the household survey.** (DOCX)

**S2 Table. Logistic regression analyses of association between socio-demographic variables and consumption of ultra-processed and deep-fried foods.** (DOCX)

## Acknowledgments

We thank Shams El Arifeen, Senior Director, Maternal and Child Health, icddr,b, for his continued support. We are immensely thankful to the MINIMat adolescents and their mothers for their participation.

## Author Contributions

**Conceptualization:** Mohammad Redwanul Islam, Eva-Charlotte Ekström.

**Data curation:** Mohammad Redwanul Islam, Syed Moshfiqur Rahman, Md. Monjur Rahman, Jesmin Pervin, Anisur Rahman.

**Formal analysis:** Mohammad Redwanul Islam.

**Funding acquisition:** Anisur Rahman, Eva-Charlotte Ekström.

**Investigation:** Syed Moshfiqur Rahman, Md. Monjur Rahman, Jesmin Pervin, Anisur Rahman, Eva-Charlotte Ekström.

**Methodology:** Mohammad Redwanul Islam, Eva-Charlotte Ekström.

**Project administration:** Syed Moshfiqur Rahman, Anisur Rahman, Eva-Charlotte Ekström.

**Supervision:** Eva-Charlotte Ekström.

**Writing – original draft:** Mohammad Redwanul Islam.

**Writing – review & editing:** Mohammad Redwanul Islam, Syed Moshfiqur Rahman, Md. Monjur Rahman, Jesmin Pervin, Anisur Rahman, Eva-Charlotte Ekström.

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
