## [Decision Letter · Decision Letter 0]

9 Mar 2022

PONE-D-21-27454Consumption of ultra-processed and deep-fried foods among rural adolescents: a cross-sectional study from BangladeshPLOS ONE

Dear Dr. Islam,

Thank you for submitting your manuscript to PLOS ONE. After careful consideration, we feel that it has merit but does not fully meet PLOS ONE’s publication criteria as it currently stands. Therefore, we invite you to submit a revised version of the manuscript that addresses the points raised during the review process.

We look forward to receiving your revised manuscript.

Kind regards,

Ammal Mokhtar Metwally, Ph.D (MD)

Academic Editor

PLOS ONE

Journal Requirements:

Additional Editor Comments (if provided):

Please note that your manuscript was reviewed by 4 experts in the field. There is consensus agreement that the idea of the article is interesting. Meanwhile, some of the reviewers identified problems in your submission and provided copious comments. Please consider responding to the reviewers’ remarks. The manuscript could be greatly strengthened by considering editing according to the specific mentioned comments.

Reviewers' comments:

Reviewer's Responses to Questions

**Comments to the Author**

1. Is the manuscript technically sound, and do the data support the conclusions?

Reviewer #1: Yes

Reviewer #2: Yes

Reviewer #3: Yes

Reviewer #4: Yes

2. Has the statistical analysis been performed appropriately and rigorously? 

Reviewer #1: Yes

Reviewer #2: Yes

Reviewer #3: Yes

Reviewer #4: Yes

3. Have the authors made all data underlying the findings in their manuscript fully available?

Reviewer #1: Yes

Reviewer #2: Yes

Reviewer #3: Yes

Reviewer #4: Yes

4. Is the manuscript presented in an intelligible fashion and written in standard English?

Reviewer #1: Yes

Reviewer #2: Yes

Reviewer #3: Yes

Reviewer #4: Yes

5. Review Comments to the Author

Reviewer #1: General comment

I am interesting for this research title. The manuscript is well prepared.

specific comment

In the introduction part

Include the magnitude of the problem from global to specific( consumption of UPF and deep fried food)

and also add efforts made to reduce this consumption

Methods and material part:

L# 129 : for operational definition of consumption measurement required reference.

Discussion part: you are try to discuss your result, but the discussion part lack clarity and need comparison from other study.

therefore try to search other study and compare it.

First discuss your outcome variables with other findings and the clinical implication of your result

the discuss the independent variables that have significant association with separate paragraph with clinical implication.

Reviewer #2: Apart from line 163 which needs some very minor revision as there is a grammatical error in the use of words. The rest of the manuscript is well written, easy to read and very informative. It gives valuable insight of what is becoming a huge problem in low and middle income countries.

Reviewer #3: Reviewer’s Comments

Journal: PLOS ONE

Title: Consumption of ultra-processed and deep-fried foods among rural adolescents: a cross-sectional study from Bangladesh.

Manuscript Number: PONE-D-21-27454

ABSTRACT

Background:

If you already assessed only gender and socio-economic factors, why you modify your title accordingly.

Methods:

Even if you write the full term, better if you remove the acronyms/abbreviation from the abstract part

Why you used 24-hour qualitative recall to assess the consumption of ultra-processed food? i.e generalizability issue by assessing it using 24-hour recall?( Why you did not use 3 days,7 days… Consumption of ultra-processed foods )

You wrote ‘Proportion of consumption was calculated and compared between groups’. Who/what were these groups must be clearly stated.

Your methods section lacks the procedures how you get the data, design you used, tools you used, how to select the study participants (techniques you used to select 2463 participants), how you analyzed etc.

Results:

Line 37: what means by ‘at least one ultra-processed or deep-fried food’? This must be clearly stated

Line 38, 40,44: Abbreviation/acronym is not recommended in abstract part

Line 40-44: Better if you write ‘aOR’ as ‘AOR’ ( Unless you have a standard in your country/institution research writing format)

Line 40: what means by strongest association here?

Line 42: you said belonging to richest household…. Did you performed/calculated Wealth Index? If so, why did not include in your methods section?

Line 43: you wrote ‘Adolescents with higher educational attainments had lower odds of consuming SSB’ but in your background section, you said ‘aimed to describe ultra-processed and deep-fried food consumption among rural adolescents by gender and socio-economic factors’. How???

Conclusions

Your conclusion lacks recommendation part

INTRODUCTION

Better if you begin your introduction section with definitions of your outcome variable

Line 52-56: needs citation

Line 57-61: needs citation

Better if you include the magnitude of malnutrition among adolescents, and from this magnitude how many of it is attributed by consumption of ultra-processed and deep fried foods

Better if you include to the end of your introduction section if there is/are national and international organizations, programs, strategies etc. which are working on adolescent nutrition to reduce this nutritional problem (your outcome variable), its achievement, gaps, what was the reason for its failures in achieving their goals (if any) to show your justification (gaps you fill) why you did this research.

METHODS

Line 103-104: put how many (n=) eligible adolescents you tracked.

Line 104-105: What did you do if the mother/s was/were died during this period (2001/ 2003 up to 2017/2019) especially for the factors directly related to maternal side that may not be addressed by asking guardians?

Line 106-110: better if you put this to the first part your ‘Participants, data collection and study site”

Line 130: Why you used 24 hour recall method to assess the consumption of ultra-processed and deep fried foods?

Line 131-132: What was your standard/reference that ‘consumption of roughly one tablespoonful or more of one or more items qualified the participant as consumer’?

Line 138-139: Factors you stated here vs you stated in your abstract part is inconsistent. Why?

Line 140-145: Please clearly show how you computed wealth index in your PCA i.e steps you pass through to reach final classifications of wealth index.

Line 146-147: What means by years here’ primary (1-5 years), and secondary (6-12 years) for adolescents or secondary 147 and above (≥ 6 years)’???

Line 158-159: better if you write the actual values you got for your model adequacy test.

RESULTS

Line 169-170: is writing this sentence/information in ‘RESULTS’ section appropriate? Better if you write this in your ‘METHODS’ section.

Line 170: What you conclude about your result with response rate 75.45%?

Line 174: you did not mention any thing about anthropometric measurements in your Methods as well as Abstract section. But you stated here about it. Why this is so?

Line 177: What means by ‘mothers attained similar or higher educational status’?

Line 182: Before if you give subtitle before stating about table 2.

Line 183-184: better if you include the time frame that when 83% of adolescents consumed these foods i.e is within 24hrs,3 day …

Line 185-193: Include the CI for your prevalence finding you get for each you state here.

Line 216-217: Why did not show its odds ratio?

DISCUSSION

Better if you begin your discussion part by stating some introduction part.

Line 227-231: What is the importance of stating these in discussion part? It is repetition of your result.

Before discussing with factors , you must discuss first your prevalence finding(consumption prevalence/proportion) with other studies done which were done on similar titles with yours ( stating findings which were in line with your finding, stating findings which prevalence were lower or higher from your findings by including possible reason why this difference(lower or higher) could exist).

Line 275-277: What is the importance of stating this unless you included in your document and showed what results you found regarding to this issue.

In discussing the significant factors of your finding, please discuss all the significant factors one by one with giving strong justification.

Line 298: Is it a limitation i.e did not know his before conducting it?

Line 302-303: delete one preposition ‘to”

CONCLUSION

Line 315-318: Please make specific your recommendation and also based on your findings.

Reviewer #4: This is an interesting study and the authors have analysed in rural adolescent consumption of ultra-processed and deep-fried foods: a cross-sectional study from Bangladesh collected a unique dataset using cutting edge methodology. The paper is generally well written and structured. However, in my opinion the paper has some shortcomings in regards to some data analyses and text, and I feel this unique dataset has not been utilized to its full extent. The methods were explained in detailed and systematic manner. The corresponding results were discussed in an interactive way. The paper written well with clear conclusions. Beyond the potentiality of the manuscript, I have a concern that the authors may address. I recommended for publication in "PLOS ONE" if following minor corrections are addressed.

1.In introduction page 3, line no 52-56 “Suboptimal diets typically have a higher share of refined sugar, added salt and trans-fat with a decreasing proportion of whole grains, fruits and vegetables, nuts and seeds. These attributes are related with a global shift away from diets based on homemade meals to those dominated by energy-dense, highly processed food and drink products”. Provide any relevant reference for the statement.

2. Page 6, line 111 the word “fired” should be “fried” it seems.

3. Any specific reason or outcome of the research for the higher consumption pattern by boys/men compared to girls/women?

3. Page 11, line no 229 Any specific reason or outcome of the research for the higher consumption pattern by boys/men compared to girls/women?

4. Conclusion of the MS has to be explained.

6. PLOS authors have the option to publish the peer review history of their article (what does this mean?). If published, this will include your full peer review and any attached files.

Reviewer #1: No

Reviewer #2: **Yes: **Patricia Mupeta Bobo

Reviewer #3: No

Reviewer #4: **Yes: **Dr Manne Munikumar

---

## [Author Response · Author response to Decision Letter 0]

17 May 2022

Reviewer #1

General comment

I am interesting for this research title. The manuscript is well prepared.

specific comment

In the introduction part

Include the magnitude of the problem from global to specific( consumption of UPF and deep fried food)

and also add efforts made to reduce this consumption

Response

We have added the followings in the manuscript:

• Consumption of UPFs has escalated across the world with their dietary share reaching as high as 57% of the daily energy intake in some settings [5]. After dominating food systems in high-income countries, availability of mass-produced UPFs is rapidly growing in low- and middle-income countries (LMICs) [6]. (Lines 64–67)

• Nonetheless, policy responses to minimize consumption of these unhealthy foods remain patchy in LMICs [25]. (Lines 80–81)

Methods and material part:

L# 129 : for operational definition of consumption measurement required reference.

Response

We have added reference for 24-hour dietary recall as a method for dietary assessment in epidemiologic studies (reference number 40, line 136). 

Discussion part: you are try to discuss your result, but the discussion part lack clarity and need comparison from other study.

therefore try to search other study and compare it.

First discuss your outcome variables with other findings and the clinical implication of your result

the discuss the independent variables that have significant association with separate paragraph with clinical implication.

Response

We appreciate this feedback and acknowledge that it is important to compare and contrast our findings with those of existing studies in the Discussion section. However, we would like to mention that the degree of comparability depends on several factors: context and setting of the studies, socio-demographic characteristics of the study populations, design and methods adopted in the studies and so forth. We explored socio-economic and gender stratification of ultra-processed and deep-fried food consumption in a cohort of rural adolescents from Bangladesh–a lower-middle-income country. Therefore, the most appropriate comparisons could be made with studies investigating socio-economic factors and gender dimension relevant to consumption among adolescents from similar rural settings in low- and middle-income countries (LMICs). Interestingly, there appears to be a notable paucity of studies on rural adolescents’ consumption of ultra-processed and deep-fried foods in LMICs. Studies focusing on ultra-processed food consumption are mostly from high-income and upper-middle-income countries [1–9]. This is also a critical research gap given the widening availability of these unhealthy foods in low-income and lower-middle-income countries [10–12] and our findings address this gap to some extent. Moreover, we did not use food frequency questionnaire. The qualitative recall in our study precluded quantification of the caloric contribution of ultra-processed foods in daily energy intake. This has limited the comparability with studies [1,3,8,9] that described the caloric contribution of ultra-processed foods. Within this scope of comparability, we have referred to findings from other studies in relation to: prevalence of sugar-sweetened beverage and “instant” food consumption (lines 267–274), association of socio-economic status with consumption of these two groups of ultra-processed foods (lines 278–280), gendered consumption of ultra-processed foods (lines 291–293), and association of educational status with ultra-processed food consumption (please see lines 299–304). We would also like to mention that reflecting on clinical implications of findings from a descriptive-analytical study on gender and socio-economic stratification could be unconvincing and may extend beyond the scope of the study.

Reviewer #2

Apart from line 163 which needs some very minor revision as there is a grammatical error in the use of words. The rest of the manuscript is well written, easy to read and very informative. It gives valuable insight of what is becoming a huge problem in low and middle income countries.

Response

We have corrected the sentence from “This study was conducted in strict accordance with according to the Declaration of Helsinki.” to “This study was conducted in strict accordance with the Declaration of Helsinki.”. Please see line 172.

Reviewer #3

ABSTRACT

Background:

If you already assessed only gender and socio-economic factors, why you modify your title accordingly.

Response

We have revised the Title to “Gender and socio-economic stratification of ultra-processed and deep-fried food consumption among rural adolescents: a cross-sectional study from Bangladesh” (lines 1–2).

Methods:

Even if you write the full term, better if you remove the acronyms/abbreviation from the abstract part

Response

We have removed “UPF” (line 31). However, removing “SSB” will make the Abstract longer than 300 words, particularly while accommodating the other changes suggested.

Why you used 24-hour qualitative recall to assess the consumption of ultra-processed food? i.e generalizability issue by assessing it using 24-hour recall?( Why you did not use 3 days,7 days… Consumption of ultra-processed foods )

Response

Dietary assessment is a challenging aspect of nutritional epidemiology. There are four traditional, subjective methods of dietary assessment employed in nutritional epidemiology: 24-hour recall, food-frequency questionnaire, dietary record, and dietary history [13]. We employed a 24-hour recall strategy in line with the objective of population-level assessment. This method offers the advantages of simplicity, cost-effectiveness and low respondent burden. In order to minimize the recall bias in this study, interviewers followed a structured approach: initial qualitative recall, followed by probing with pictorial charts, followed by an interactive summarization. This is quite similar to multiple-pass approaches successfully employed in large-scale surveys elsewhere [14,15]. While there are limitations inherent to the 24-hour recall just like there are limitations in the other three methods [13], validation studies suggest 24-hour recall to be reliable at population level [16,17].

You wrote ‘Proportion of consumption was calculated and compared between groups’. Who/what were these groups must be clearly stated.

Response

We have revised the sentence, now it reads: “Proportion of consumption was calculated and compared by gender and household wealth” (line 34).

Your methods section lacks the procedures how you get the data, design you used, tools you used, how to select the study participants (techniques you used to select 2463 participants), how you analyzed etc.

Response

We have included the study design (line 29). The collection of data (household survey), method (24-hour recall) and analysis (bivariate comparison and logistic regression) have been mentioned. The participants come from a birth cohort called the MINIMat cohort [18] and details have been outlined in the manuscript (lines 178–182).

Results:

Line 37: what means by ‘at least one ultra-processed or deep-fried food’? This must be clearly stated

Response

This indicates the percentage of adolescents who reportedly consumed at least one item from any of the five food groups (four groups of ultra-processed foods and one group of deep-fried foods). Considering the word limit, we replaced the word “food” with “item” in that sentence for clarity (line 39).

Line 38, 40,44: Abbreviation/acronym is not recommended in abstract part

Response

We removed one (line 31), the remaining help with the adherence to the word limit of 300 for Abstract.

Line 40-44: Better if you write ‘aOR’ as ‘AOR’ ( Unless you have a standard in your country/institution research writing format)

Response

We have replaced “aOR with “AOR” (lines 41–45).

Line 40: what means by strongest association here?

Response

This implies to the magnitude of association in terms of how large the adjusted odds ratio for consumption of sugar-sweetened beverage is compared to the adjusted odds ratios for consumption of the other three groups of ultra-processed foods and of the deep-fried foods. The adjusted odds ratio for sugar-sweetened beverage was the highest among the five groups.

Line 42: you said belonging to richest household…. Did you performed/calculated Wealth Index? If so, why did not include in your methods section?

Response

We calculated asset scores using principal component analysis, which has been mentioned in the main Methods section (lines 146–151). In the Abstract, we added: “Asset scores were constructed” (line 33). With this addition the Abstract now stands at 300 words, the prescribed limit.

Line 43: you wrote ‘Adolescents with higher educational attainments had lower odds of consuming SSB’ but in your background section, you said ‘aimed to describe ultra-processed and deep-fried food consumption among rural adolescents by gender and socio-economic factors’. How???

Response

Both maternal and adolescents’ education plays an important role in determining the socio-economic status/position of a household [19]. Use of these two variables in epidemiological studies is commonplace. Simply put, socio-economic status/position relates not only to income but also to educational attainment of the family members, financial security (linked with education as well) and perceptions of social status and class [19,20]. Therefore, we considered these two variables as part and indicative of “socio-economic factors”.

Conclusions

Your conclusion lacks recommendation part

Response

We acknowledge that it is important to outline a recommendation in the Conclusion part of the Abstract. Considering the word limit, we reiterated our principal findings in this section of the Abstract in relation to the study aim mentioned in the Background of the Abstract: “... to describe ultra-processed and deep-fried food consumption ... by gender and socio-economic factors, and to explore their relative importance in shaping consumption.”. Some may also suggest that proposing recommendation for public health initiative or policymaking based on a single, descriptive-analytical study of cross-sectional design could be erroneous. However, our findings are of relevance in improving the understanding of the gender and socio-economic dimensions of consumption of unhealthy foods among rural adolescents in LMICs. Accordingly, we pointed out that the gendered consumption may heighten the risk of diet-related non-communicable diseases among the boys (line 49–50).

INTRODUCTION

Better if you begin your introduction section with definitions of your outcome variable

Response

We began the Introduction section by putting into perspective the global burden of suboptimal diet in terms of morbidity (reflected by disability-adjusted life-years) and mortality at population level. The key features of suboptimal diet–a greater share of refined sugar, added salt and trans-fat¬–are linked to consumption of ultra-processed and deep-fried foods, the subject matter of our study. After the short first paragraph, we introduced the term of ultra-processed food and presented its definition (lines 58–64).

Line 52-56: needs citation

Response

The data on morbidity and mortality burden of suboptimal diet mentioned come from the Global Burden of Disease Study 2017. The related article by Afshin et al. has been cited (reference number 1, line 53). The reference for the sentence “Suboptimal diets typically have a higher share of refined sugar, added salt and trans-fat with a decreasing proportion of whole grains, fruits and vegetables, nuts and seeds” has been added (reference number 2, line 55). The reference for the sentence “These attributes are related with a global shift away from diets based on homemade meals to those dominated by energy-dense, highly processed food and drink products” has been added (reference number 3, line 57).

Line 57-61: needs citation

Response

The required citation has been given (reference numbers 3–5, line 64).

Better if you include the magnitude of malnutrition among adolescents, and from this magnitude how many of it is attributed by consumption of ultra-processed and deep fried foods

Response

We could not find a study specifically quantifying how much of the “magnitude” of malnutrition can be “attributed” to consumption of ultra-processed and deep-fried foods. Malnutrition encompasses both over- and under-nutrition. We mentioned the presence of malnutrition in LMIC context: “Stunting, thinness, and food insecurity affect a significant proportion of adolescents in these settings [28]. Consequently, access to inexpensive calories through these foods may expose them to a disproportionately high NCD risk in ensuing adulthood [29,30].” (lines 86–89). We also mentioned that “While trend of adolescent overweight and obesity has plateaued in high-income countries, it has accelerated in South Asia” (lines 92–93). This change in trajectory paralleled the increasing sales of ultra-processed foods in LMICs. 

Better if you include to the end of your introduction section if there is/are national and international organizations, programs, strategies etc. which are working on adolescent nutrition to reduce this nutritional problem (your outcome variable), its achievement, gaps, what was the reason for its failures in achieving their goals (if any) to show your justification (gaps you fill) why you did this research.

Response

We would like to mention that the outcome variable in the study is consumption of ultra-processed and deep-fried foods. We mentioned in the Introduction that a greater consumption of these foods has been linked to a range of nutritional outcomes (lines 70–71). However, “nutritional problem” per se was not the outcome variable that we assessed. We aimed to describe the dietary consumption of ultra-processed and deep-fried foods among a group of rural adolescents and to explore the association of gender and socio-economic factors with consumption. Whereas the overarching goal is to generate evidence that could inform measures aiming to reduce consumption of unhealthy foods among adolescents, the specific objective of the study centered on capturing the gender and socio-economic dimensions (lines 95–99). We consider an outline of activities undertaken by different organizations working on “adolescent nutrition” to go beyond the scope of this piece of empirical research.

METHODS

Line 103-104: put how many (n=) eligible adolescents you tracked.

Response

We have added the number as suggested (line 109). Moreover, details regarding this has been mentioned at the beginning of the Results section: “Adolescents born to MINIMat mothers as singletons with valid birth anthropometrics were eligible to participate. Out of 3267 eligible adolescents, 2465 (75.45%) completed the household survey of 15-year follow-up. The reasons for loss to follow-up were: outmigration (n = 656), child death (n = 94), and refusal to participate (n = 52).” (lines 178–181).

Line 104-105: What did you do if the mother/s was/were died during this period (2001/ 2003 up to 2017/2019) especially for the factors directly related to maternal side that may not be addressed by asking guardians?

Response

In the rare (less than 1%) instance of mother not being present, the woman (usually grandmother or aunt to the adolescent) in charge of cooking and preparing food for the family was the respondent. The information collected during the household survey that we used in this study were well within the grasp of such a respondent. 

Line 106-110: better if you put this to the first part your ‘Participants, data collection and study site”

Response

The first subheading in the Methods section is Participants, data collection and study site (line 101). We wrote the subsection according to the sequence suggested in this subheading: first, we mentioned the study design and source of study participants; then, we briefly outlined the timespan and mode of data collection; finally, we wrote about the study site of Matlab to give readers an impression of the setting. 

Line 130: Why you used 24 hour recall method to assess the consumption of ultra-processed and deep fried foods?

Response

As detailed above, we employed 24-hour recall for assessment of dietary consumption because of the simplicity, cost-effectiveness, and reliability at population level that this method offers in the context of a considerably large-scale survey. This method poses low respondent burden as well. Like other methods of dietary assessment used in nutritional epidemiology, the 24-hour recall also has caveats [13]. We acknowledged the pertinent limitations in the Discussion section (lines 307–315). Furthermore, recent validation studies suggest the performance of 24-hour recall to be satisfactory [16,17].

Line 131-132: What was your standard/reference that ‘consumption of roughly one tablespoonful or more of one or more items qualified the participant as consumer’?

Response

The trained interviewers were provided with tablespoons similar to those commonly used in Matlab. They displayed the tablespoons while ascertaining consumption during the household survey [21]. One tablespoonful amount corresponds roughly to 15 grams. We wanted to ensure that trivial consumption (less than 15 grams) does not classify a participant as a consumer. This has been recommended by researchers with expertise in adopting 24-hour recall method in LMIC settings [22].

Line 138-139: Factors you stated here vs you stated in your abstract part is inconsistent. Why?

Response

The lines 138–139 in the submitted manuscript reads “Socio-demographic variables included in the analysis were gender, household wealth, maternal and adolescents’ education.”. Here, the notion “socio-demographic” can be viewed as a contraction of “socio-economic” and “demographic”. The latter includes gender; it is well established that gender is a demographic variable [23,24]. Regarding “socio-economic”, as elaborated above, we posit that–household wealth, educational status of mothers and of adolescents–all three are socio-economic factors/variables [19,20]. Therefore, we opine that there is no contradiction with what we have written in the Abstract as aim of the study: “... aimed to describe ultra-processed and deep-fried food consumption among rural adolescents by gender and socio-economic factors, and to explore their relative importance in shaping consumption.” (lines 25–27).

Line 140-145: Please clearly show how you computed wealth index in your PCA i.e steps you pass through to reach final

Response

We used principal component analysis (PCA) for constructing an asset score for each household. This approach has been widely employed in LMICs to create a variable that would capture the construct of socio-economic status [25–27]. PCA is a multivariate data reduction technique that reduces the number of variables in a data set into a smaller number of ‘dimensions’ or ‘components’ where each component is a linear weighted combination of the initial variables [27]. The steps that we followed are similar to those implemented by The DHS Program in their nationally representative Demographic and Health Surveys conducted in LMICs. These steps have been described in detail elsewhere [25,27]. Briefly, the steps are as follows:

1. Collection of data on asset variables 

We collected data on i. ownership of a set of durable assets (e.g., mobile phone, radio, television, refrigerator, bicycle, etcetera), ii. access to electricity and sanitary latrine, and ii. nature of fuel used. These are context-specific “economic proxy” variables [27].

2. Recoding categorical data and application of PCA

The output from a PCA contains factor coefficient scores for each variable and weights (eigenvectors) for the principal components. The eigenvector for each principal component reflects the percentage of variation in the data explained by that principal component. The first principal component is the one that explains the largest amount of variation.

3. Creating the asset score

Using the factor scores for variables from the first principal component as weights, a dependent variable was constructed for each household. This variable–the asset score–has a mean of zero and a standard deviation of one. The higher the score, the higher the implied socio-economic status of that household relative to the setting of Matlab.

4. Categorizing the households into tertiles

Asset scores arranged in ascending order were divided into tertiles with the lowest, intermediate and highest representing the poorest, the middle-status and the richest households, respectively.

We included the following in the manuscript: “PCA yields factor scores for each variable and eigenvectors for the principal components. The asset score variable, with a mean of zero and a standard deviation of one, was created using the factor scores for variables within the first principal component as weights.” (lines 149–151). We have also added the reference: line 146 (reference number 41).

Line 146-147: What means by years here’ primary (1-5 years), and secondary (6-12 years) for adolescents or secondary 147 and above (≥ 6 years)’???

Response

We categorized educational status of mothers and adolescents according to the number of years of formal education they have attained. In Bangladesh, primary education spans 1–5 years (up to 5th grade) and secondary education spans 6–12 years (up to 12th grade). For the adolescents, it was not possible to attain education beyond secondary level at the time of the survey as they have been followed up at 15 years. For example, if a participant starts formal education at 5 years of age (which is usual in rural Bangladesh), it is not possible for her/him to receive more than 12 years of formal education at the age of 15 years. However, because of their higher age, it was possible for mothers to pursue education beyond secondary level and some of them did attain education beyond secondary level. That is why, for mothers, the category was “Secondary and above (≥ 6y)”.

Line 158-159: better if you write the actual values you got for your model adequacy test.

Response

A p-value greater than 0.05 obtained from the Hosmer-Lemeshow Test is indicative of an adequate fit [28]. The p-values for the five models (corresponding to the five food groups) have been reported in the last row of Table 3 (0.876, 0.941, 0.349, 0.945, and 0.715). All these values are greater than 0.05.

RESULTS

Line 169-170: is writing this sentence/information in ‘RESULTS’ section appropriate? Better if you write this in your ‘METHODS’ section.

Response

According to the STROBE (STrengthening the Reporting of OBservational studies in Epidemiology) Checklist (can be found here: https://www.strobe-statement.org/checklists/), Item 13, these information regarding number of eligible participants, number of participants included, reasons for non-participation, etcetera should be reported in the Results section. We followed this recommendation.

Line 170: What you conclude about your result with response rate 75.45%?

Response

We consider this a reasonably high retention rate given the rural context of a lower-middle-income country and the long, 15-year span of the follow-up. We would like to leave the conclusion to be drawn regarding this percentage up to the learned readers.

Line 174: you did not mention any thing about anthropometric measurements in your Methods as well as Abstract section. But you stated here about it. Why this is so?

Response

The study reported in the manuscript was a part of the 15-year follow-up of MINIMat trial (lines 102–110). Anthropometric data were collected in the 15-year follow-up. As anthropometry is not directly related to the research question that this study addressed, we did not mention the details of anthropometric data collection in the Methods section. We provided the descriptive statistics for height and weight in Table 1 to give readers a coherent impression of the sample characteristics. 

Line 177: What means by ‘mothers attained similar or higher educational status’?

Response

As elaborated above, this means that 44.6% of the mothers attained secondary education or higher than secondary education. The word “similar” refers back to the words “secondary education” in the previous line.

Line 182: Before if you give subtitle before stating about table 2.

Response

We would like to keep the Results section as one continuous, uninterrupted segment in the manuscript as it is not extensively large.

Line 183-184: better if you include the time frame that when 83% of adolescents consumed these foods i.e is within 24hrs,3 day …

Response

We have added the “timeframe”, now the sentence reads “About 83% of the adolescents reportedly consumed at least one ultra-processed or deep-fried food in the past 24 hours” (lines 192–193)

Line 185-193: Include the CI for your prevalence finding you get for each you state here.

Response

We included the confidence intervals for confectionery and sugar-sweetened beverage, the most and the least consumed groups, respectively (lines 194–195). Confidence interval around each of the proportions has been provided in Table 2. Mentioning all these confidence intervals again in the body of the text would be a repetition that we wanted to avoid.

Line 216-217: Why did not show its odds ratio?

Response

We did not present the odds ratios here as these were not statistically significant. However, the odds ratios with 95% confidence intervals can be found in S2 Table (Logistic regression analyses of association between socio-demographic variables and consumption of ultra-processed and deep-fried foods) as we mentioned in the last sentence of the of the Results section: “The complete regression analyses with both the crude and adjusted ORs can be found in S2 Table” (lines 231–232).

DISCUSSION

Better if you begin your discussion part by stating some introduction part.

Response

In the first paragraph of the Discussion section, we concisely presented the main findings in relation to the stated aim of the study. This is a commonly recommended practice [29], and also supported by PLOS [30]. 

Line 227-231: What is the importance of stating these in discussion part? It is repetition of your result.

Response

As mentioned immediately above, we reiterated the principal findings in the first paragraph of the Discussion section to provide an overview of what we found.

Before discussing with factors, you must discuss first your prevalence finding(consumption prevalence/proportion) with other studies done which were done on similar titles with yours ( stating findings which were in line with your finding, stating findings which prevalence were lower or higher from your findings by including possible reason why this difference(lower or higher) could exist).

Response

While we acknowledge this important feedback, we would like to mention that we noted a paucity of studies focusing on ultra-processed and deep-fried food consumption among rural adolescents in low- and lower-middle-income countries; more so when it comes to South and South-East Asia. However, we still compared and contrasted our findings with comparable existing studies to the best of our knowledge: context-specific aspects regarding sources/sellers of packaged ultra-processed foods in Matlab (lines 245–246), prevalence of sugar-sweetened beverage consumption (lines 267–272), low prevalence of consumption of ready-to-eat items (lines 272–274), association of household wealth with consumption of ready-to-eat foods and sugar-sweetened beverage (lines 275–279), association of gender with consumption (lines 291–296), and association of educational status with consumption (lines 299–304). 

Line 275-277: What is the importance of stating this unless you included in your document and showed what results you found regarding to this issue.

Response

Here, we referred to our previous work that described a diet that is limited in diversity and dominated by starchy staple with lower consumption of fruits, vegetables and animal-source foods in the same sample. This is indicative of low diet quality. Consumption of ultra-processed foods along with a low-quality, nutrient-deficient diet has previously been observed in other resource-limited settings of LMICs [12,31]. This combination can be labelled as a “double burden of suboptimal diet” and may establish an unhealthy diet pattern among rural adolescents [12]. We inferred that health implications of such a diet pattern need to be studied further (lines 289–290).

In discussing the significant factors of your finding, please discuss all the significant factors one by one with giving strong justification.

Response

We broadly adhered to this suggestion in the Discussion section: the first paragraph gives an overview of the main findings; the second paragraph deals with the two most prevalent food groups (packaged confectioneries and deep-fried foods) along with probable context-specific reasons for the high prevalences observed and its implication; the third paragraph deals with the relatively low consumption of sugar-sweetened beverage and ready-to-eat foods, and probable reasons for association of household wealth with consumption of these two groups but not the other three; the fourth paragraph briefly contextualizes the findings in relation to published studies from Matlab and similar rural settings; the fifth paragraph discusses the association of gender with consumption and association of educational status with consumption of sugar-sweetened beverage; and the sixth paragraph addresses limitations of the study. 

Line 298: Is it a limitation i.e did not know his before conducting it?

Response

We omitted the sentence “The cross-sectional design precluded assessment of causality” (line 308).

Line 302-303: delete one preposition ‘to”

Response

We omitted one “to” (line 313).

CONCLUSION

Line 315-318: Please make specific your recommendation and also based on your findings.

Response

We added the following: “Public health initiatives for promoting healthy diet among adolescents should take into account the gendered consumption of unhealthy foods” (lines 325–326). We omitted the following to avoid any inference not directly based on the findings: “Further studies are needed to characterize the food environment in Matlab and how adolescents interact with this rural food environment” (lines 328–330)

Reviewer #4

This is an interesting study and the authors have analysed in rural adolescent consumption of ultra-processed and deep-fried foods: a cross-sectional study from Bangladesh collected a unique dataset using cutting edge methodology. The paper is generally well written and structured. However, in my opinion the paper has some shortcomings in regard to some data analyses and text, and I feel this unique dataset has not been utilized to its full extent. The methods were explained in detailed and systematic manner. The corresponding results were discussed in an interactive way. The paper written well with clear conclusions. Beyond the potentiality of the manuscript, I have a concern that the authors may address. I recommended for publication in "PLOS ONE" if following minor corrections are addressed.

1.In introduction page 3, line no 52-56 “Suboptimal diets typically have a higher share of refined sugar, added salt and trans-fat with a decreasing proportion of whole grains, fruits and vegetables, nuts and seeds. These attributes are related with a global shift away from diets based on homemade meals to those dominated by energy-dense, highly processed food and drink products”. Provide any relevant reference for the statement.

Response

The reference for the sentence “Suboptimal diets typically have a higher share of refined sugar, added salt and trans-fat with a decreasing proportion of whole grains, fruits and vegetables, nuts and seeds” has been added (reference number 2, line 55). The reference for the sentence “These attributes are related with a global shift away from diets based on homemade meals to those dominated by energy-dense, highly processed food and drink products” has been added (reference number 3, line 57).

2. Page 6, line 111 the word “fired” should be “fried” it seems.

Response

We corrected the misspelled word (line 117).

3. Any specific reason or outcome of the research for the higher consumption pattern by boys/men compared to girls/women?

Response

The association of gender with consumption of ultra-processed food appears to vary across settings. As we noted in lines 291–293, some studies demonstrate a significantly greater consumption among men/boys [3,5], while other studies report the opposite [9,32]. When it comes to consumption of sugar-sweetened beverage (which includes energy drinks as well), the male predominance appears to be clearer [33–35]. The reasons for these heterogeneities remain poorly understood. Moreover, we need further studies to pinpoint the reason for gendered consumption of ultra-processed and deep-fried foods among adolescents in Matlab.

3. Page 11, line no 229 Any specific reason or outcome of the research for the higher consumption pattern by boys/men compared to girls/women?

Response

We reflected on the likely reasons in the 5th paragraph of the Discussion section (line 296–298). Of note, the likely reasons–taste preference differing by gender, greater access of boys to retailers or vendors selling these foods, or intentional avoidance by girls because of body image concerns–could not be examined further as we did not collect data on these aspects.

4. Conclusion of the MS has to be explained.

Response

We have revised the Conclusion section. We added the following: “Public health initiatives for promoting healthy diet among adolescents should take into account the gendered consumption of unhealthy foods” (lines 325–326). We omitted the following to avoid any inference not directly based on the findings: “Further studies are needed to characterize the food environment in Matlab and how adolescents interact with this rural food environment” (lines 328–330). To reiterate, we aimed to describe the gender and socio-economic stratification. We found a pre-eminent role of gender among Matlab adolescents, whereas household wealth influenced consumption of two of the four groups of ultra-processed foods (ready-to-eat items and sugar-sweetened beverage).

Reference:

1. Vandevijvere S, De Ridder K, Fiolet T, Bel S, Tafforeau J. Consumption of ultra-processed food products and diet quality among children, adolescents and adults in Belgium. European Journal of Nutrition. 2018 [cited 20 Jan 2019]. doi:10.1007/s00394-018-1870-3

2. Juul F, Martinez-Steele E, Parekh N, Monteiro CA, Chang VW. Ultra-processed food consumption and excess weight among US adults. British Journal of Nutrition. 2018;120: 90–100. doi:10.1017/S0007114518001046

3. Moubarac J-C, Batal M, Louzada ML, Martinez Steele E, Monteiro CA. Consumption of ultra-processed foods predicts diet quality in Canada. Appetite. 2017;108: 512–520. doi:10.1016/j.appet.2016.11.006

4. Martínez Steele E, Popkin BM, Swinburn B, Monteiro CA. The share of ultra-processed foods and the overall nutritional quality of diets in the US: evidence from a nationally representative cross-sectional study. Population Health Metrics. 2017;15. doi:10.1186/s12963-017-0119-3

5. Rauber F, Steele EM, Louzada ML da C, Millett C, Monteiro CA, Levy RB. Ultra-processed food consumption and indicators of obesity in the United Kingdom population (2008-2016). PLoS ONE. 2020;15: e0232676. doi:https://doi.org/10.1371/journal.pone.0232676

6. Srour B, Fezeu LK, Kesse-Guyot E, Allès B, Méjean C, Andrianasolo RM, et al. Ultra-processed food intake and risk of cardiovascular disease: prospective cohort study (NutriNet-Santé). BMJ. 2019;365: l1451. doi:10.1136/bmj.l1451

7. D’Avila HF, Kirsten VR. Energy intake from ultra-processed foods among adolescents. Rev Paul Pediatr. 2017;35: 54–60. doi:10.1590/1984-0462/;2017;35;1;00001

8. Cediel G, Reyes M, da Costa Louzada ML, Martinez Steele E, Monteiro CA, Corvalán C, et al. Ultra-processed foods and added sugars in the Chilean diet (2010). Public Health Nutrition. 2018;21: 125–133. doi:10.1017/S1368980017001161

9. Bielemann RM, Motta JVS, Minten GC, Horta BL, Gigante DP. Consumption of ultra-processed foods and their impact on the diet of young adults. Revista de Saúde Pública. 2015;49: 1–10. doi:10.1590/S0034-8910.2015049005572

10. Baker P, Friel S. Food systems transformations, ultra-processed food markets and the nutrition transition in Asia. Global Health. 2016;12: 80. doi:10.1186/s12992-016-0223-3

11. Vandevijvere S, Jaacks LM, Monteiro CA, Moubarac J-C, Girling‐Butcher M, Lee AC, et al. Global trends in ultraprocessed food and drink product sales and their association with adult body mass index trajectories. Obesity Reviews. 2019;20: 10–19. doi:10.1111/obr.12860

12. Ochola S, Masibo PK. Dietary Intake of Schoolchildren and Adolescents in Developing Countries. Annals of Nutrition and Metabolism. 2014;64: 24–40. doi:10.1159/000365125

13. Shim J-S, Oh K, Kim HC. Dietary assessment methods in epidemiologic studies. Epidemiol Health. 2014;36: 1–8. doi:10.4178/epih/e2014009

14. Raper N, Perloff B, Ingwersen L, Steinfeldt L, Anand J. An overview of USDA’s Dietary Intake Data System. Journal of Food Composition and Analysis. 2004;17: 545–555. doi:10.1016/j.jfca.2004.02.013

15. Pries AM, Rehman AM, Filteau S, Sharma N, Upadhyay A, Ferguson EL. Unhealthy Snack Food and Beverage Consumption Is Associated with Lower Dietary Adequacy and Length-for-Age z-Scores among 12–23-Month-Olds in Kathmandu Valley, Nepal. The Journal of Nutrition. 2019;149: 1843–1851. doi:10.1093/jn/nxz140

16. Arsenault JE, Moursi M, Olney DK, Becquey E, Ganaba R. Validation of 24‐h dietary recall for estimating nutrient intakes and adequacy in adolescents in Burkina Faso. Matern Child Nutr. 2020;16: e13014. doi:10.1111/mcn.13014

17. Nguyen PH, Huybregts L, Sanghvi TG, Tran LM, Frongillo EA, Menon P, et al. Dietary Diversity Predicts the Adequacy of Micronutrient Intake in Pregnant Adolescent Girls and Women in Bangladesh, but Use of the 5-Group Cutoff Poorly Identifies Individuals with Inadequate Intake. The Journal of Nutrition. 2018;148: 790–797. doi:10.1093/jn/nxy045

18. Arifeen SE, Ekström E-C, Frongillo EA, Hamadani J, Khan AI, Naved RT, et al. Cohort Profile: The Maternal and Infant Nutrition Interventions in the Matlab (MINIMat) Cohort in Bangladesh. International Journal of Epidemiology. 2018 [cited 9 Aug 2018]. doi:10.1093/ije/dyy102

19. Galobardes B. Indicators of socioeconomic position (part 1). Journal of Epidemiology & Community Health. 2006;60: 7–12. doi:10.1136/jech.2004.023531

20. Herd P, Goesling B, House JS. Socioeconomic Position and Health: The Differential Effects of Education versus Income on the Onset versus Progression of Health Problems. J Health Soc Behav. 2007;48: 223–238. doi:10.1177/002214650704800302

21. Islam MR, Rahman SM, Tarafder C, Rahman MdM, Rahman A, Ekström E-C. Exploring Rural Adolescents’ Dietary Diversity and Its Socioeconomic Correlates: A Cross-Sectional Study from Matlab, Bangladesh. Nutrients. 2020;12: 2230. doi:10.3390/nu12082230

22. Arimond M, Wiesmann D, Becquey E, Carriquiry A, Daniels MC, Deitchler M, et al. Simple Food Group Diversity Indicators Predict Micronutrient Adequacy of Women’s Diets in 5 Diverse, Resource-Poor Settings. J Nutr. 2010;140: 2059S-2069S. doi:10.3945/jn.110.123414

23. Demographics. Encyclopedia of Research Design. California, US: SAGE Publications, Inc.; 2010. doi:10.4135/9781412961288.n108

24. Martin P, OSMOND MW, HESSELBART S, Wood MP. The Significance of Gender as a Social and Demographic Correlate of Sex Role Attitudes. Sociological Focus. 1980;13: 383–396. 

25. Rutstein SO, Johnson K. The DHS Wealth Index. DHS Comparative Reports No. 6. Maryland, USA: ORC Macro; 2004 p. 71. Available: https://dhsprogram.com/pubs/pdf/cr6/cr6.pdf

26. Gwatkin DR, Rutstein S, Johnson K, Suliman E, Wagstaff A, Amouzou A. Socio-Economic Differences in Health, Nutrition, and Population Within Developing Countries: An Overview. Washington, DC: The World Bank; 2007 Oct p. 287. Report No.: 48361. Available: http://documents.worldbank.org/curated/en/962091468332070548/

27. Vyas S, Kumaranayake L. Constructing socio-economic status indices: how to use principal components analysis. Health Policy and Planning. 2006;21: 459–468. doi:10.1093/heapol/czl029

28. Hosmer DW, Lemeshow S, Sturdivant RX. Applied Logistic Regression. 3rd ed. New Jersey: John Wiley & Sons, Incorporated; 2013. 

29. Cals JWL, Kotz D. Effective writing and publishing scientific papers, part VI: discussion. Journal of Clinical Epidemiology. 2013;66: 1064. doi:10.1016/j.jclinepi.2013.04.017

30. PLOS. How to Write Discussions and Conclusions. In: Resources [Internet]. [cited 17 Mar 2022]. Available: https://plos.org/resource/how-to-write-conclusions/

31. Contreras M, Blandón EZ, Persson L-Å, Hjern A, Ekström E-C. Socio-economic resources, young child feeding practices, consumption of highly processed snacks and sugar-sweetened beverages: a population-based survey in rural northwestern Nicaragua. BMC Public Health. 2015;15. doi:10.1186/s12889-015-1374-5

32. Monteles N, Santos O K, Gomes K, Pacheco R M, Gonçalves F K. The impact of consumption of ultra-processed foods on the nutritional status of adolescents. Rev chil nutr. 2019;46: 429–435. doi:10.4067/S0717-75182019000400429

33. Cunha DB, da Costa THM, da Veiga GV, Pereira RA, Sichieri R. Ultra-processed food consumption and adiposity trajectories in a Brazilian cohort of adolescents: ELANA study. Nutr & Diabetes. 2018;8: 28. doi:10.1038/s41387-018-0043-z

34. Sim E, Sohn W, Choi E-S, Noh H. Sugar-sweetened beverage consumption frequency in Korean adolescents: based on the 2015 Youth Risk Behavior Web-Based Survey. International Dental Journal. 2019;69: 376–382. doi:https://doi.org/10.1111/idj.12485

35. Gui Z-H, Zhu Y-N, Cai L, Sun F-H, Ma Y-H, Jing J, et al. Sugar-Sweetened Beverage Consumption and Risks of Obesity and Hypertension in Chinese Children and Adolescents: A National Cross-Sectional Analysis. Nutrients. 2017;9. doi:10.3390/nu9121302

---

## [Decision Letter · Decision Letter 1]

22 Jun 2022

PONE-D-21-27454R1Gender and socio-economic stratification of ultra-processed and deep-fried food consumption among rural adolescents: a cross-sectional study from BangladeshPLOS ONE

Dear Dr. Islam,

Thank you for submitting your manuscript to PLOS ONE. After careful consideration, we feel that it has merit but does not fully meet PLOS ONE’s publication criteria as it currently stands. Therefore, we invite you to submit a revised version of the manuscript that addresses the points raised during the review process.

We look forward to receiving your revised manuscript.

Kind regards,

Ammal Mokhtar Metwally, Ph.D (MD)

Academic Editor

PLOS ONE

Journal Requirements:

Additional Editor Comments:

Great effort was made by the authors to utilize the feedback that was provided for them to correct their manuscript. I find it interesting and improved with respect to the original submission. Please consider responding to the reviewers’ remarks. The manuscript could be greatly strengthened by considering editing according to the specific mentioned comments

Reviewers' comments:

Reviewer's Responses to Questions

**Comments to the Author**

1. If the authors have adequately addressed your comments raised in a previous round of review and you feel that this manuscript is now acceptable for publication, you may indicate that here to bypass the “Comments to the Author” section, enter your conflict of interest statement in the “Confidential to Editor” section, and submit your "Accept" recommendation.

Reviewer #1: All comments have been addressed

Reviewer #3: All comments have been addressed

2. Is the manuscript technically sound, and do the data support the conclusions?

Reviewer #1: Yes

Reviewer #3: Yes

3. Has the statistical analysis been performed appropriately and rigorously? 

Reviewer #1: Yes

Reviewer #3: Yes

4. Have the authors made all data underlying the findings in their manuscript fully available?

Reviewer #1: Yes

Reviewer #3: Yes

5. Is the manuscript presented in an intelligible fashion and written in standard English?

Reviewer #1: No

Reviewer #3: Yes

6. Review Comments to the Author

Reviewer #1: Abstract

Avoid abbreviation from abstract

Introduction part

First start with the definition of your outcome variable clearly

When you write the magnitude of the problem be specific on your study population or among adolescents “It accounted for an estimated 11 million deaths and 53 255 million disability-adjusted life-years (DALYs) in 2017” this sentence is not clear among adolescent or the general population?

The second paragraph is very long from L#58 –L#81 divided into two or three paragraphs to make easy for reader

In the last paragraph the gap you mentioned not clear?

Methods

In L# 137 -138 “Reported consumption of roughly one tablespoonful or more of one or more items qualified the participant as consumer of that particular group of ultra-processed or deep-fried food.” Is there any reference?

Result

In table 1 for categorical variable (for example gender, educational status, source of drinking water) why you report mean and SD better to report frequency and percentage?

Discussion part

Your discussion part lack coherence, so, need revision

Better to start the discussion with your objective then explain your result with the first paragraph

Then compare your finding with other study findings like consistence with, higher or lower than after that for the difference write possible explanation and clinical implication of your finding

Why you discuss the significantly associated factors of UPF and deep fired food?

For example gender, wealth quintile

In the conclusion part: conclude based on your finding for example ‘Household wealth was associated with the consumption of ready-to-eat or “instant” foods 222 and SSB”

Reviewer #3: The paper focuses on very interesting topic, very important segment of population (Adolescents) and uses large datasets. I am happy on this and I personal recommend it for acceptance of publication. As a comment, please try to address some grammatical, editorial issues, and in some part; clarity of some statements (for easily understandability for readers) will be checked and addressed.

7. PLOS authors have the option to publish the peer review history of their article (what does this mean?). If published, this will include your full peer review and any attached files.

Reviewer #1: No

Reviewer #3: No

---

## [Author Response · Author response to Decision Letter 1]

23 Jun 2022

Reviewer #1

Abstract

Avoid abbreviation from abstract

Response

We have omitted the abbreviation for sugar-sweetened beverage (i.e., SSB) and used the full form throughout the Abstract. (Lines 32, 39–40, 41 and 46)

Introduction part

First start with the definition of your outcome variable clearly

Response

We have revised accordingly and added the following at the beginning of the Introduction: “Ultra-processed foods (UPFs) refer to multi-ingredient, industrial formulations composed of such food-derived substances as starch, fat, oil, sugar, casein, etcetera; or synthesized through complex processing of food constituents like whey, gluten, soya protein isolate, maltodextrin, and corn syrup [1,2]”. (Lines 53–56)

When you write the magnitude of the problem be specific on your study population or among adolescents “It accounted for an estimated 11 million deaths and 53 255 million disability-adjusted life-years (DALYs) in 2017” this sentence is not clear among adolescent or the general population?

Response

We have added “among adults” for clarification. (Line 58)

The second paragraph is very long from L#58 –L#81 divided into two or three paragraphs to make easy for reader

Response

As suggested, we have divided this paragraph into two shorter paragraphs: one dealing with the increasing consumption of ultra-processed foods (Lines 63–76), and another dealing with the problem posed by deep-fried foods (Lines 77–86).

In the last paragraph the gap you mentioned not clear?

Response

In the last paragraph of the Introduction, we highlighted the issue that adolescents in low- and middle-income countries (LMICs) now have a greater access to inexpensive calories, while the quality of their diet remains limited (Lines 88–97). In parallel to this, the trend of overweight and obesity among adolescents has accelerated in South Asian LMICs, but the trend remained stable in high-income countries (Lines 97–98). This observation is reported in a large, pooled analysis by Abarca-Gómez and colleagues (manuscript Reference number 35) that was published in The Lancet in 2017 [1]. Thereafter, we pointed out that adolescents in Bangladesh are also facing a similar situation (Lines 98–100). Consumption of ultra-processed and deep-fried foods is a well known driver of the rising burden of overweight and obesity. However, empirical evidence on gender and socio-economic stratification of consumption of these unhealthy foods among Bangladeshi adolescents is lacking. This is the “gap” that the present study aimed to address. Context-specific understanding of the relationship of gender and socio-economic factors with consumption of these unhealthy foods is necessary for informing targeted public health initiatives.

For clarity, we revised as follows: “... Ultra-processed and deep-fried foods are known dietary drivers of increasing overweight and obesity [15,18]. Nevertheless, socio-demographic correlates of consumption of these unhealthy foods among rural, Bangladeshi adolescents remain unexplored. Hence, we aimed to describe ultra-processed and deep-fried food consumption among rural adolescents by gender and socio-economic factors, and to explore their relative importance in shaping consumption.”. (Lines 100–106)

Methods

In L# 137 -138 “Reported consumption of roughly one tablespoonful or more of one or more items qualified the participant as consumer of that particular group of ultra-processed or deep-fried food.” Is there any reference?

Response

This requirement of consumption of roughly one tablespoonful was intended for avoiding foods eaten in trivial amounts from categorizing a participant as a consumer. This has been recommended in the context of dietary assessment using food groups [2], and employed in our previous work in Matlab [3] as well.

We have now added the reference to the manuscript (Manuscript Reference number 41). (Line 145).

Result

In table 1 for categorical variable (for example gender, educational status, source of drinking water) why you report mean and SD better to report frequency and percentage?

Response

As suggested, we reported frequency with percentage for categorical variables and mean with standard deviation for numerical variables. The heading of the right column of Table 1 reads “n (%) or mean (SD)”, which indicates that for categorical variables like gender, educational status, etcetera the values represent frequency (“n” stands for frequency here) along with percentage in the parentheses. For example, in case of gender, we had 1261 girls (the frequency or n) that constitute 51.2% (the percentage in the parentheses) of the study sample. Therefore, the value 51.2 in the parentheses does not represent standard deviation, rather represents the percentage of girls in the sample.

We have added a legend to the Table 1: “The values represent frequency (n) with percentage for categorical variables and mean with standard deviation (SD) for numerical variables”. (Lines 197–198)

Discussion part

Your discussion part lack coherence, so, need revision

Better to start the discussion with your objective then explain your result with the first paragraph

Response

We have added the following at the beginning of the Discussion: “The objective of the present study was to describe and analyze the gender and socio-economic stratification of ultra-processed and deep-fried food consumption in a cohort of rural adolescents”. (Lines 242–244)

Then compare your finding with other study findings like consistence with, higher or lower than after that for the difference write possible explanation and clinical implication of your finding

Response

Regarding the structure of the Discussion section: we presented an abridged summary of the principal findings in the first paragraph (Lines 242–252).

In the second paragraph (Lines 253–277), we focused on the extent of consumption in terms of the proportion of adolescents who reported to have consumed ultra-processed and deep-fried foods. We contextualized the extent of consumption with regard to our study setting, which is a rural area (Lines 253–264). We, then, focused on the reported consumption of deep-fried foods which was fairly high and discussed the broader implications of deep-fried food consumption (Lines 264–277). Overall, this paragraph puts into perspective the described consumption of the five groups of unhealthy foods.

In the third paragraph (278–291), we focused particularly on sugar-sweetened beverage and read-to-eat or “instant” foods. Consumption of these two groups were relatively low in our sample, and thus, contrasted with findings from urban settings elsewhere (Lines 279–285). Furthermore, these are the two food groups (out of five) for which household wealth showed a positive association with consumption (Table 3 in the manuscript). Therefore, we suggested that this could indicate a potential role of purchasing ability (Lines 283–286). In the last two sentences of this paragraph, we compared our findings regarding household wealth status (i.e., positive association with consumption for only two food groups) with that of previous studies (Lines 289–291).

In the fourth paragraph (292–308), we offered a context-specific perspective on the emerging problem of ultra-processed food consumption in the rural setting of Matlab based on our previous research in the area. We have also added a brief discussion on clinical implication as suggested by the Reviewer (Lines 301–308, please see below for further clarification).

In the fifth paragraph (Lines 302–317), we highlighted the gender dimension in our findings – one of the main focuses of the study. Furthermore, we compared and contrasted the gender stratification observed in our study with that of the existing studies from elsewhere.

In the last paragraph (Lines 318–328), we mentioned pertinent limitations and methodological considerations.

We perceive that the overall structure of the Discussion section serves the purpose of highlighting the major findings in line with the stated aim of the study and contextualizing them against the backdrop of what has been reported in existing studies.

Discussing clinical implications of consuming ultra-processed and deep-fried foods within the scope of a cross-sectional study among adolescents as clinical outcomes manifest often in ensuing adulthood. However, in line with the suggestion, we added the following: “Consumption of ultra-processed and deep-fried foods has been associated with several clinical outcomes including metabolic syndrome, hypertension, dyslipidemia and coronary artery disease [25,61]. These associations were observed predominantly among adults, yet the underlying cardio-metabolic alterations may commence during adolescence [62]. Besides, the habit of consuming unhealthy foods established during adolescence can track into adulthood [63]. Accordingly, the gender stratification observed in this study might play a role in exposing the boys to a higher cardiometabolic risk in upcoming adulthood”. (Lines 301–308) 

Why you discuss the significantly associated factors of UPF and deep fired food? For example gender, wealth quintile

Response

We highlighted significantly associated factors in the Discussion for two reasons: (i) their direct relevance to the stated aim of the study, which is “Hence, we aimed to describe ultra-processed and deep-fried food consumption among rural adolescents by gender and socio-economic factors, and to explore their relative importance in shaping consumption” (Lines 104–106); and (ii) it would be erroneous to draw inference from results (particularly odds ratios) that were not statistically significant.

Our aim was to investigate the gender and socio-economic stratification. We found that gender was significantly associated with consumption for all five food groups whereby the odds of consumption were significantly higher among boys than girls. On the other hand, odds of consumption were significantly higher among adolescents from the richest households only for two food groups: ready-to-eat or “instant” foods and sugar-sweetened beverage. We drew inferences accordingly.

In the conclusion part: conclude based on your finding for example ‘Household wealth was associated with the consumption of ready-to-eat or “instant” foods 222 and SSB”

Response

As suggested, we have added the following: “Household wealth was associated with consumption of ready-to-eat or “instant” foods and SSB, but not the other three food groups. This may indicate a role of purchasing ability in the rural setting” in the Conclusion section. (Lines 342–344)

Reviewer #3

The paper focuses on very interesting topic, very important segment of population (Adolescents) and uses large datasets. I am happy on this and I personal recommend it for acceptance of publication. As a comment, please try to address some grammatical, editorial issues, and in some part; clarity of some statements (for easily understandability for readers) will be checked and addressed.

Response

We greatly appreciate the comments from the Reviewer. We have revised and made adjustments for further clarity. (Lines 100–104, 242–244, 301–308).

References

1. Abarca-Gómez L, Abdeen ZA, Hamid ZA, Abu-Rmeileh NM, Acosta-Cazares B, Acuin C, et al. Worldwide trends in body-mass index, underweight, overweight, and obesity from 1975 to 2016: a pooled analysis of 2416 population-based measurement studies in 128·9 million children, adolescents, and adults. The Lancet. 2017;390: 2627–2642. doi:10.1016/S0140-6736(17)32129-3

2. Arimond M, Wiesmann D, Becquey E, Carriquiry A, Daniels MC, Deitchler M, et al. Simple Food Group Diversity Indicators Predict Micronutrient Adequacy of Women’s Diets in 5 Diverse, Resource-Poor Settings. J Nutr. 2010;140: 2059S-2069S. doi:10.3945/jn.110.123414

3. Islam MR, Rahman SM, Tarafder C, Rahman Md M, Rahman A, Ekström E-C. Exploring Rural Adolescents’ Dietary Diversity and Its Socioeconomic Correlates: A Cross-Sectional Study from Matlab, Bangladesh. Nutrients. 2020;12: 2230. doi:10.3390/nu12082230

---

## [Decision Letter · Decision Letter 2]

18 Jul 2022

Gender and socio-economic stratification of ultra-processed and deep-fried food consumption among rural adolescents: a cross-sectional study from Bangladesh

PONE-D-21-27454R2

Dear Dr. Islam,

We’re pleased to inform you that your manuscript has been judged scientifically suitable for publication and will be formally accepted for publication once it meets all outstanding technical requirements.

Kind regards,

George Vousden

Staff Editor

PLOS ONE

Additional Editor Comments (optional):

Reviewers' comments:

Reviewer's Responses to Questions

**Comments to the Author**

1. If the authors have adequately addressed your comments raised in a previous round of review and you feel that this manuscript is now acceptable for publication, you may indicate that here to bypass the “Comments to the Author” section, enter your conflict of interest statement in the “Confidential to Editor” section, and submit your "Accept" recommendation.

Reviewer #1: All comments have been addressed

2. Is the manuscript technically sound, and do the data support the conclusions?

Reviewer #1: Yes

3. Has the statistical analysis been performed appropriately and rigorously? 

Reviewer #1: Yes

4. Have the authors made all data underlying the findings in their manuscript fully available?

Reviewer #1: Yes

5. Is the manuscript presented in an intelligible fashion and written in standard English?

Reviewer #1: Yes

6. Review Comments to the Author

Reviewer #1: This paper is very interesting, the authors address all my comments properly, therefore I recommend this paper should be accepted for publication.

7. PLOS authors have the option to publish the peer review history of their article (what does this mean?). If published, this will include your full peer review and any attached files.

Reviewer #1: No

---

## [Editor Report · Acceptance letter]

20 Jul 2022

PONE-D-21-27454R2 

Gender and socio-economic stratification of ultra-processed and deep-fried food consumption among rural adolescents: a cross-sectional study from Bangladesh 

Dear Dr. Islam:

I'm pleased to inform you that your manuscript has been deemed suitable for publication in PLOS ONE. Congratulations! Your manuscript is now with our production department. 

Kind regards, 

on behalf of

Dr. George Vousden 

Staff Editor

PLOS ONE